# Mitigation of Cadmium Toxicity through Modulation of the Frontline Cellular Stress Response

**Soisungwan Satarug [1,*], David A. Vesey [1,2]** and **Glenda C. Gobe [1,3,4]**

1    Kidney Disease Research Collaborative, Translational Research Institute, Brisbane, QLD 4102, Australia
2    Department of Nephrology, Princess Alexandra Hospital, Brisbane, QLD 4102, Australia
3    School of Biomedical Sciences, The University of Queensland, Brisbane, QLD 4072, Australia
4    NHMRC Centre of Research Excellence for CKD QLD, UQ Health Sciences,
     Royal Brisbane and Women's Hospital, Brisbane, QLD 4029, Australia
*    Correspondence: sj.satarug@yahoo.com.au

**Abstract:** Cadmium (Cd) is an environmental toxicant of public health significance worldwide. Diet is the main Cd exposure source in the non-occupationally exposed and non-smoking populations. Metal transporters for iron (Fe), zinc (Zn), calcium (Ca), and manganese (Mn) are involved in the assimilation and distribution of Cd to cells throughout the body. Due to an extremely slow elimination rate, most Cd is retained by cells, where it exerts toxicity through its interaction with sulfur-containing ligands, notably the thiol (-SH) functional group of cysteine, glutathione, and many Zn-dependent enzymes and transcription factors. The simultaneous induction of heme oxygenase-1 and the metal-binding protein metallothionein by Cd adversely affected the cellular redox state and caused the dysregulation of Fe, Zn, and copper. Experimental data indicate that Cd causes mitochondrial dysfunction via disrupting the metal homeostasis of this organelle. The present review focuses on the adverse metabolic outcomes of chronic exposure to low-dose Cd. Current epidemiologic data indicate that chronic exposure to Cd raises the risk of type 2 diabetes by several mechanisms, such as increased oxidative stress, inflammation, adipose tissue dysfunction, increased insulin resistance, and dysregulated cellular intermediary metabolism. The cellular stress response mechanisms involving the catabolism of heme, mediated by heme oxygenase-1 and -2 (HO-1 and HO-2), may mitigate the cytotoxicity of Cd. The products of their physiologic heme degradation, bilirubin and carbon monoxide, have antioxidative, anti-inflammatory, and anti-apoptotic properties.

**Keywords:** bilirubin; cadmium; carbon monoxide; glycolysis; gluconeogenesis; heme; heme oxygenase-1; heme oxygense-2; obese phenotype; heme oxygenase-2 deficiency; stress response

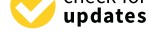



## 1. Introduction

The utility of a redox inert metal cadmium (Cd) in many industrial processes, and the use of phosphate fertilizers contaminated with Cd by the agricultural sector have resulted in widespread dispersion of this toxic metal in the environment and subsequently the food chains [1–5]. Volcanic emissions, biomass and fossil fuel combustion, and cigarette smoke are additional sources of environmental Cd pollution [6–10]. Cd in cigarette smoke as a volatile metallic form and oxide (CdO) has a particularly high transmission rate [10,11]. An existence of the nose-to-brain transport route of toxic metals raises the possibility that airborne Cd may enter the central nervous system (CNS) by utilizing a Cd-altered blood–brain barrier [12–14].

Foods that are frequently consumed in large quantities, such as rice, potatoes, wheat, leafy salad vegetables, and other cereal crops, form the most significant dietary sources of Cd [15–17]. Seafood (shellfish), mollusks, and crustaceans are additional dietary Cd sources [18,19]. Cd enters the body from the gut and lungs via the metal transporters and pathways for zinc (Zn), calcium (Ca), iron (Fe), manganese (Mn), and possibly selenium

(Se) [17]. Because of an extremely slow excretion rate, most absorbed Cd is retained in cells, and the cellular content of Cd increases with the duration of exposure (age) [17].

Evidence from epidemiologic and experimental studies suggest that low environmental exposure to Cd may increase the risk of diseases with high prevalence, such as chronic kidney disease (CKD), liver disease, type 2 diabetes, and neurodegenerative disorders [15–17,20]. Developing strategies to prevent these chronic ailments is of global importance in the absence of effective chelation therapies to reduce the Cd body burden.

In the present review, we focus on the effects of Cd exposure on cellular intermediary metabolism and the cytoprotective role of metal-induced stress responses. Toxic manifestation of Cd in kidneys, liver, pancreas, and adipose tissues are discussed because these organs are central to the control of blood glucose levels. Blood and urinary Cd levels that were found to be associated with adverse metabolic outcomes are provided. Evidence for Cd-induced oxidative stress and inflammatory conditions are reviewed. The interplays of heme oxygenase-1 and-2 (HO-1 and HO-) to regulate glycolysis and gluconeogenesis are highlighted as is their key role as the frontline cellular stress response mechanism that neutralizes oxidative damage and protects against abnormal glucose metabolism, excessive weight gain, and obesity.

## 2. Measures of Human Cadmium Exposure

### 2.1. Entry, Distribuion, and Excretion of Cadmium

As Figure 1 depicts, ingested Cd is absorbed by the intestine and transported via the portal blood system to liver, while inhaled Cd is transported to lungs and possibly the CNS via nasal-to-brain route. Cd induces synthesis of metallothionein (MT) and the CdMT complexes are formed in these organs [21,22].

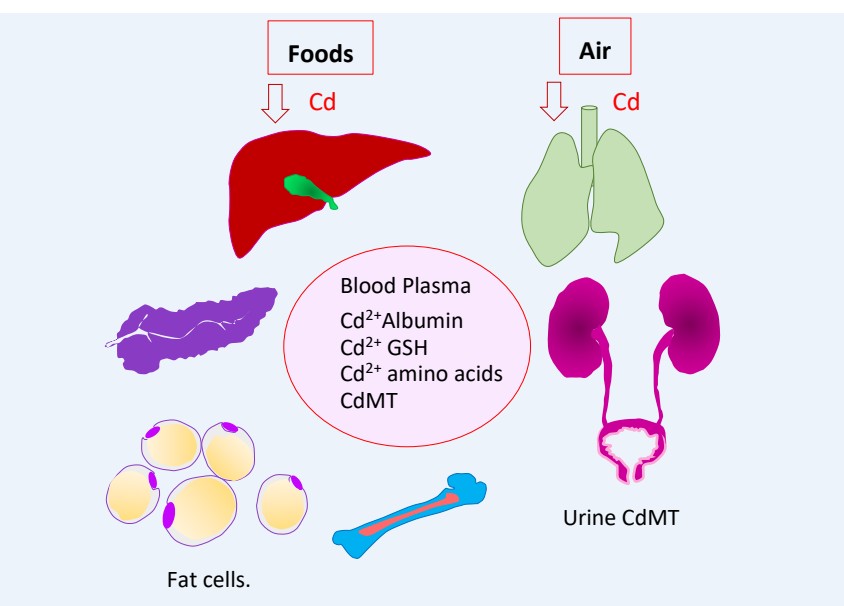

**Figure 1.** Entry, distribution, and exit of cadmium. Ingested $Cd^{2+}$ ions are absorbed and transported to liver, while inhaled Cd oxide and metallic Cd are transported to lungs. The fraction of absorbed $Cd^{2+}$ ions not taken up by hepatocytes in the first pass reaches systemic circulation and is assimilated by cells throughout the body. In liver and lungs, Cd induces synthesis of MT with resultant formation of CdMT with subsequent release into the circulation and reabsorption by kidney tubular cells. The Cd absorbed by the gut and lungs is eventually accumulated in the kidney tubular cells and is excreted in urine as CdMT by injured or dying tubular epithelial cells of the kidneys.

The fraction of absorbed Cd not taken up by hepatocytes in the first pass reaches the systemic circulation and is taken up by tissues and organs throughout the body, including the adipose tissue [23,24], pancreas [25], lungs, liver, and kidneys [26]. All nucleated cells

have the capacity to assimilate $Cd^{2+}$ ions that are not bound to MT through the transporters for the metals required for normal cellular metabolism and function.

Most cells, hepatocytes included, do not assimilate CdMT due to a lack of mechanisms for protein internalization (endocytosis). Kidney tubular epithelial cells are well equipped with such mechanisms, which facilitate reabsorption of virtually all filtered proteins for reutilization [27,28]. Kidney tubular cells also assimilate Cd in non-MT forms through many other kidney tubular transporter systems.

## 2.2. Endogenous Suppliers of Cadmium-Metallothionein Complexes

The cellular formation of CdMT has been viewed as a detoxification mechanism that prevents acute toxicity because the "free" $Cd^{2+}$ ion is the chemically reactive toxic form of this metal. In theory, each MT molecule can carry 7 atoms of $Cd^{2+}$, 7 atoms of $Zn^{2+}$ or 12 atoms of $Cu^{2+}$, and the complexes are denoted as $Cd_7MT$, $Zn_7MT$, and $Cu_{12}MT$ [29]. However, various species of mixed metal complexes, such as $Cd_3Cu_3ZnMT$, $Cd_4CuZn_2MT$, and $Cd_6CuMT$, are formed in vivo, with the molar contents of Cd dependent on levels of Cd exposure [29].

There are at least 16 MT isoforms, and they belong to four major families, MT-1– MT-4 [30]. Among these families, MT-1 and MT-2 are the most frequently expressed in tissues, including leucocytes and kidney tubular epithelial cells [31–33]. MT-3 has a higher binding affinity for Cu than MT-1/2, and it is expressed in high abundance, particularly in kidneys and neurons [34–36]. Cu bound to MT-3 may be involved in the nephrotoxicity and neurotoxicity of Cd because Cu is a redox active metal that can cause oxidative stress [35].

The $Cd^{2+}$ ions sequestered in hepatic CdMT complexes are those from the diet while pulmonary CdMT contains airborne Cd. Liver and lungs serve as endogenous reservoirs from which CdMT complexes are released as cells die. CdMT complexes are redistributed to kidneys. Although the formation of CdMT complexes prevent acute cytotoxicity, it may increase the risk of long-term toxicity because $Cd^{2+}$ ions can be released under certain conditions, leading to an increased synthesis of nitric oxide (NO) that liberates the $Cd^{2+}$ ions, previously bound to MT [37–39].

## 2.3. Blood Cadmium as an Indicator of Recent Exposure

In the circulation, less than 10% of Cd is present in plasma, and the remainder is in erythrocytes, where most Cd in whole blood is found. The chloride/bicarbonate anion exchanger ($[Cl^-/HCO_3^-]$) is responsible for Cd uptake by erythrocytes [40]. In plasma, Cd is bound to the amino acid histidine and proteins, such as MT, pre-albumin, albumin, $\alpha$2-macroglobulin, and immunoglobulins G and A [41–43]. At any given time, the whole blood Cd level is indicative of recent exposure because the average lifespan of erythrocytes is 120 days. The biological half-life of blood Cd ranged between 75 and 128 days [44].

## 2.4. Urinary Cadmium as an Indicator of Cumulative Lifetime Exposure

The kidney burden of Cd as µg/g tissue weight increases with age proportionally to the amount assimilated from exogenous sources over a lifetime [26,45–47]. The biological half-life of Cd in the kidney cortex was estimated at 30 years for non-smokers [38,39,45,46]. Urine Cd has long been used as an indicator of a cumulative lifetime exposure because this parameter is correlated with the kidney burden of Cd and other determinants of absorption rate that include the body status of Fe and Zn [26,48]. However, the excretion of Cd is indicative of injury to tubular epithelial cells of the kidneys, discussed in Section 2.4. Interestingly, a study of environmentally exposed Chinese subjects aged 2.8 to 86.8 years (*n* = 1235) showed that Cd excretion levels increased with age, peaking at 50 years in non-smoking women and 60 years in non-smoking men [49].

## 2.5. Roles for Zinc Transporters in the Biliary Excretion and Cytotoxicity of Cadmium

A Swedish autopsy study reported that approximately 0.001–0.005% of Cd in the body was excreted in urine each day [45,46]. This kidney route of Cd excretion is extremely

slow. In comparison, the biliary excretion rate of Cd appeared to be higher and it was suggested that bile might be an important excretion route for Cd [46]. The effects of GSH and dithiothreitol on Cd uptake and on biliary release of Cd were demonstrated using the isolated perfused rat liver [50].

The biliary route of Cd excretion has gained support from recent research data showing that ZIP8, a zinc transporter, was involved in hepatic excretion of Mn through bile [51]. Because ZIP8 also mediated Cd transport, it remains to be seen if ZIP8 mediates biliary Cd excretion [51,52]. It is relevant that the expression of the ZIP8 gene was modulated by intracellular glutathione (GSH) concentrations [53], and the hepatotoxicity of Cd in rats could be attenuated by GSH administration [54].

Evidence for Zn influx and Zn efflux transporters, notably ZIP8, ZIP14, and ZnT1, as the determinants of Cd cytotoxicity is increasing [55–58]. A few are summarized herein. The pretreatment of the rat liver epithelial cells (TRL1215) with cyproterone, a synthetic steroidal antiandrogen with a structure related to progesterone, decreased sensitivity to Cd through a reduction in Cd accumulation [59]. However, the molecular basis for such a decrease in Cd accumulation was not investigated. It was shown in another study that silencing the expression of a Zn/Cd efflux transporter, ZnT1 resulted in an increased Cd accumulation and enhanced Cd toxicity [60]. A decrease in Cd accumulation together with a decrease in ZIP8 expression, assessed by ZIP8 mRNA and ZIP8 protein levels, was seen in metallothionein-null cells that were resistant to Cd toxicity [61,62].

### 2.6. Urine Cadmium as a Warning Sign of Toxicity in Progress

It has long been viewed that excreted Cd included Cd molecules that pass through the glomerular filtration membrane into the filtrate but are not reabsorbed [63]. However, it is noteworthy that the principal form of Cd in urine is CdMT [64], and that the excreted CdMT originates from injured or dying tubular cells [65]. Thus, Cd excretion is a manifestation of the cytotoxicity of Cd accumulation in kidneys' tubular cells even at very low exposure levels. Our conceptual framework accounting for the pathogenesis of Cd-induced nephropathy originating from tubular cell injury is depicted in Figure 2.

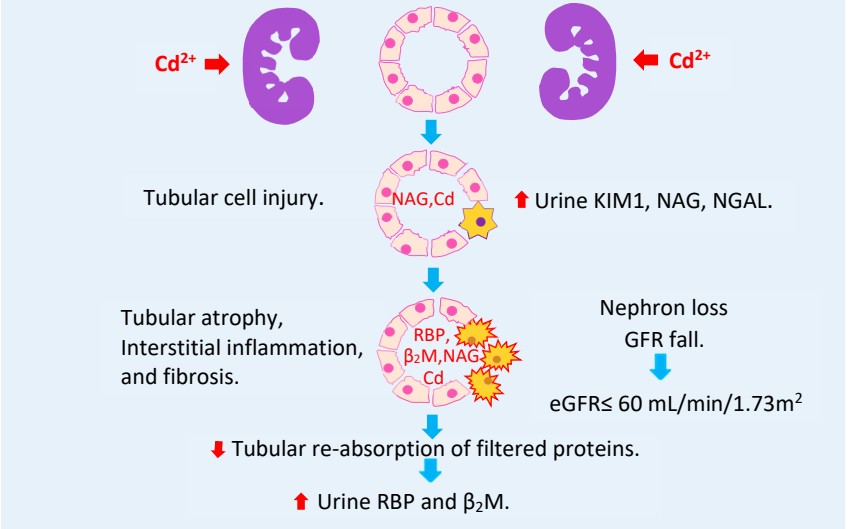

**Figure 2.** Sequential outcomes of tubular cell toxic injury of cadmium accumulation in kidneys. Cd inflicts tubular cell injury at low intracellular concentrations, and the toxicity intensifies as Cd concentration rises [65]. Tubular injury disables glomerular filtration, leading to nephron atrophy, glomerulosclerosis, and interstitial inflammation and fibrosis. A reduction in tubular reabsorption of filtered proteins, RBP and β2MG follows tubular atrophy and nephron loss. Abbreviation: KIM1, kidney injury molecule 1; NAG, N-acetyl-β-D-glucosaminidase, NGAL, neutrophil gelatinase associated lipocalin; RBP, retinal binding protein; $\beta_2M$, $\beta_2$-microglobulin.

In a histopathological examination of kidney biopsies from healthy kidney transplant donors [66], the degree of tubular atrophy was positively associated with the level of Cd accumulation. Tubular atrophy was observed at relatively low Cd levels [66]. In Japanese residents of a Cd pollution area, the average half-life of the metal among those with lower body burden (urinary Cd < 5 μg/L) was 23.4 years; in those with higher body burden (urinary Cd > 5 μg/L), the average half-life was 12.4 years [67,68]. Thus, the lower the body burden, the longer the half-life of Cd. A half-life of 45 years was estimated from a Cd-toxicokinetic model that used data from Swedish kidney transplant donors exposed to low environmental levels [69].

## 3. Manifestation of Cadmium Toxicity

Population-based studies in many countries and the U.S. general population study known as National Health and Nutrition Examination Survey (NHANES) suggest adverse effects of chronic exposure to Cd extend beyond kidneys and bones. Table 1 provides epidemiologic evidence for the effects of Cd in organs involved in the maintenance of glucose homeostasis, including the liver [70–72], pancreas [73–75], and kidneys [76–80], In the post absorptive state, kidney and liver supply an equal amount of glucose into the systemic blood circulation [81–83].

**Table 1.** Adverse health effects of cadmium exposure in multiple organs evident from the U.S. NHANES datasets.

| Organs | NHANES Datasets | Adverse Effects and Risk Estimates | References |
|---|---|---|---|
| Liver | 1988–1994 *n* 12,732, ≥20 yrs | In women, liver inflammation was associated with urinary Cd levels ≥ 0.83 μg/g creatinine (OR 1.26). In men, liver inflammation, NAFLD and NASH were associated with urinary Cd ≥ 0.65 μg/g creatinine with respective OR values of 2.21, 1.30, and 1.95. | Hyder et al., 2013 [70] |
| Liver | 1999–2015 *n* 11, 838, ≥20 yrs | Elevated plasma ALT and AST was associated with a 10-fold increment of urinary Cd with respective OR values of 1.36 and 1.31. | Hong et al., 2021 [71] |
| Liver | 1999–2016 *n* 4411 adolescents | Elevated plasma ALT and AST were associated with urinary Cd quartile 4 with respective OR values of 1.40 and 1.64. The effect was larger in boys than girls. | Xu et al., 2022 [72] |
| Pancreas | 1988–1994 *n* 8722, ≥40 yrs | Risks of prediabetes and diabetes were associated with urinary Cd levels 1–2 μg/g creatinine with respective OR values of 1.48 and 1.24. | Schwartz et al., 2003 [73] |
| Pancreas | 2005–2010 *n* 2398, ≥40 yrs | An increased risk of prediabetes was associated with urinary Cd levels ≥ 0.7 μg/g creatinine after adjustment for covariates. | Wallia et al., 2014 [74] |
| Pancreas | 1999–2006 *n* 4530 adults | $BMDL_5$ and $BMDL_{10}$ of urinary Cd levels derived from diabetes endpoint were of 0.198 and 0.365 μg/g creatinine, respectively. | Shi et al., 2021 [75] |
| Kidneys | 1999–2006 *n* 14,778, aged ≥ 20 yrs | Reduced GFR [a] (OR 1.32), albuminuria [b] (OR 1.92), and reduced GFR plus albuminuria (OR 2.91) were associated with blood Cd levels ≥ 0.6 μg/L with respective OR values of 1.32, 1.92, and 2.91. | Navas-Acien et al., 2009 [76] |
| Kidneys | 1999–2006 *n* 5426, aged ≥ 20 yrs | Albuminuria (OR 1.63) was associated with urinary Cd levels > 1 μg/g creatinine plus blood Cd levels > 1 μg/L (OR 1.63). Reduced eGFR (OR 1.48) and albuminuria (OR 1.41) were associated with blood Cd levels > 1 μg/L with respective OR values of 1.48 and 1.41. | Ferraro et al., 2010 [77] |

**Table 1.** *Cont.*

| Organs | NHANES Datasets | Adverse Effects and Risk Estimates | References |
|---|---|---|---|
| Kidneys | 2007–2012 <br> *n* 12,577, aged $\geq$ 20 yrs | Reduced eGFR (OR 1.80) and albuminuria (OR 1.60) were associated with blood Cd levels > 0.61 µg/L with respective OR values of 1.80 and 1.60. | Madrigal et al., 2019 [78] |
| Kidneys | 2009–2012 <br> *n* 2926, aged $\geq$ 20 yrs | An elevated albumin excretion was associated with urinary Cd levels > 0.220 µg/L and blood Cd levels > 0.243 µg/L. | Zhu et al., 2019 [79] |
| Kidneys | 2011–2012 <br> *n* 1545, aged $\geq$ 20 yrs | Reduced eGFR (OR 2.21) and albuminuria (OR 2.04) were associated with blood Cd levels > 0.53 µg/L with respective OR values of 2.21 and 2.04. | Lin et al., 2014 [80] |

NHANES, National Health and Nutrition Examination Survey; *n*, sample size; HR, hazard ratio; OR, odds ratio; ALT, alanine aminotransferase; AST, aspartate aminotransferase; NAFLD, non-alcoholic fatty liver disease; NASH, non-alcoholic steatohepatitis; [a] Reduced eGFR, estimated glomerular filtration rate (eGFR) $\leq$ 60 mL/min/1.73 m$^2$; [b] Albuminuria, urinary albumin to creatinine ratio $\geq$ 30 mg/g.

The hepatoxicity of Cd was seen in both children and adults [70–72]. In adults, increases in risk of liver inflammation, NAFLD, and NASH were associated with urinary Cd levels $\geq$ 0.6 µg/g creatinine [70]. In children, hepatotoxicity of Cd was more pronounced in boys than girls [72]. In NHANES cycles undertaken between 1999 and 2016, reduced eGFR and albuminuria were consistently associated with Cd exposure measures [76–80].

### 3.1. Cadmium and the Risk of Type 2 Diabetes

Prediabetes and diabetes are defined as fasting plasma glucose $\geq$ 110 mg/dL and 126 mg/dL, respectively. The number of people with prediabetes and diabetes have reached epidemic proportions globally. The epidemic is attributed to the increasing prevalence of obesity, leading to a search for environmental obesogenic substances. In comparison, a statistically significant inverse association has consistently been observed between Cd exposure and body mass index and other measures of adiposity (Section 3.2). Dietary exposure to Cd is consequently the least expected and least recognized environmental risk factor for diabetes.

Increases in the risks of prediabetes and diabetes among NHANES 1988–1994 participants were associated with urinary Cd levels of 1–2 µg/g creatinine [73]. An increased risk of prediabetes among NHANES 2005–2010 was associated with urinary Cd levels $\geq$ 0.7 µg/g creatinine after adjustment for covariates [74]. In a risk analysis, the prevalence of type 2 diabetes was likely to be smaller than 5% and 10% at urinary Cd levels of 0.198 and 0.365 µg/g creatinine, respectively [75].

In the Wuhan-Zhuhai prospective cohort study [84], fasting blood glucose levels were found to increase with urinary Cd over a three-year observation period. For each 10-fold increase in urinary Cd, the prevalence of prediabetes rose by 42%. Dose–response relationships between Cd exposure and risks of prediabetes and diabetes were observed in two meta-analyses, [85,86]. In a risk analysis of pooled data from 42 studies, the risks of prediabetes and diabetes increased linearly with blood and urinary Cd; prediabetes risk reached a plateau at urinary Cd of 2 µg/g creatinine, and diabetes risk rose as blood Cd reached 1 µg/L [86].

### 3.2. An Inverse Relationship between Cadmium Body Burden and Obesity

The relationships between Cd exposure levels and disease shown by associative studies have often been ignored. However, it is important to recognize such associations as they may indicate mechanisms of disease pathogenesis. Thus, reports of an inverse relationship between Cd body burden and obesity provide developmental data that may lead to future significant correlations that define disease pathogenesis and aid in therapy development. Herein we report such associative studies that replicate an association observed between Cd and reduced risk of obesity. These data can be interpreted to suggest that Cd may have

caused the dysregulation of the cellular intermediary metabolism (a further discussion in Section 4.3) and that type 2 diabetes associated with Cd is independent of obesity.

Urinary Cd levels were inversely associated with central obesity among participants of NHANES 1999–2002 [87]. Among NHANES 2003–2010 participants, their blood Cd levels were inversely associated with body mass index (BMI) [88]. In another analysis of data from NHANES 2001–2014, participants aged 20–80 years (*n* = 3982), with urinary Cd levels were not associated with the risk of metabolic syndrome, but they were associated with a decreased risk of abdominal obesity [89]. In a meta-analysis of data from 11 cross-sectional studies, Cd exposure was not associated with an increased risk of metabolic syndrome, but it was associated with dyslipidemia, especially in the Asian population [90].

Urinary Cd was associated with a reduction in risk of obesity by 54% in children and adolescents enrolled in NHANES 1999–2011; an inverse association between urinary Cd and obesity was stronger in the younger age group (6–12 years) than the older age group (13–19 years) [91]. Urinary Cd levels were inversely associated with height and BMI in Flemish children, aged 14–15 years [92].

Similarly, an inverse association between blood Cd and BMI was seen in non-smokers in the Canadian Health Survey 2007–2011 [93]. A negative association between Cd exposure and various measures of obesity were seen in both men and women in a study of the indigenous population of northern Québec, Canada, where obesity was highly prevalent [94].

An inverse association between blood Cd and BMI was noted in a group of Korean men, 40–70 years of age [95]. This Korean population study observed an inverse correlation between fasting blood glucose and urinary Cd excretion levels, and a 1.81-fold increase in risk of diabetes among men who had urinary Cd > 2 µg/g creatinine.

In a Chinese study, urinary Cd excretion rates $\geq$ 2.95 µg/g creatinine were associated with reduced risk of excessive weight gain and reduced risk of obesity [96]. Higher urinary Cd levels were associated with lower BMI values in a study of residents of Shanghai without workplace exposure to Cd, showing the median urinary Cd excretion of 0.77 µg/g creatinine [97].

Of interest, lower BMI figures were associated with higher Cd accumulation levels in fat tissues in a cohort study in Spain [23]. Furthermore, an increased resistance to insulin and higher plasma insulin levels were seen in smokers whose adipose tissue Cd levels were in the middle tertile, compared to those with adipose tissue Cd levels in the lowest tertile 1 [24].

### 3.3. Cadmium-Induced Oxidative Stresss and Inflammation

The aforementioned statistically significant inverse relationship between Cd body burden and obesity suggests that an effect of Cd on the risk of diabetes is independent of adiposity and inflammation, accompanying excessive body fats. Indeed, there is evidence that Cd may cause inflammation in adipose tissues in a Swiss autopsy study, Cd accumulation in omentum visceral and abdominal subcutaneous fat tissues were quantified [98]. In an in vitro study using the adipose-derived human mesenchymal stem cells (FC-0034), Cd in the same range found in those postmortem fat tissue samples was found to disrupt cellular Zn homeostasis and to cause an increase in the expression of various pro-inflammatory cytokines [98]. Studies in mice showed that Cd caused the abnormal differentiation of adipocytes, resulting in small adipocytes and a reduction in the secretion of adiponectin [99,100].

As data in Table 2 indicate, substantial evidence for Cd-induced oxidative stress and inflammation comes from the U.S. population studies, which included NHANES III [101,102], a study of healthy New York women [103]; NHANES 2003–2010 [95]; and NHANES 1999–2002 [104,105]. The effects of Cd on the risks of cardiovascular disease and all-cause mortality were also indicated [106]. In these studies, serum γ-glutamyl transferase (GGT), C-reactive protein (CRP), and shortening of leucocyte telomere length were quantified as they were measures of increased oxidative stress and inflammation. In some of these reports, a protective role of certain nutrients was observed.

**Table 2.** Cadmium-induced oxidative stress and inflammation.

| Biomarkers | Datasets | Findings | References |
|---|---|---|---|
| Serum GGT. | NHANES III, *n* 10,098, aged ≥ 20 yrs. | Serum GGT was positively associated with urinary Cd levels between 0.002 and 23.4 µg/g creatinine. Serum vitamins C and E and carotenoids were inversely associated with GGT. | Lee et al., 2006 [101] |
| Serum CRP and fibrinogen | NHANES III, *n* 6497, aged 40–79 yrs. | Elevations of serum CRP and fibrinogen were associated with urinary Cd levels ≥ 0.93 µg/g creatinine with respective OR values of 1.24 and 2.12. | Lin et al., 2009 [102] |
| Serum bilirubin | Healthy women, Buffalo, New York *n* 259, aged 18–44 yrs. | A reduction in serum bilirubin by 4.9% was associated with a 2-fold increase in blood Cd. Median Cd level (interquartile range) was 0.3 (0.19–0.43) µg/L. | Pollack et al., 2015 [103] |
| CRP, GGT, ALP, bilirubin and white cell count. | NHANES 2003–2010, *n* 3056 women, *n* 3288 men. | Serum CRP, GGT, and ALP levels were increased, respectively, by 47.5%, 8.8% and 3.7%, in urinary Cd quartiles 4 vs. 1. Consumption of anti-oxidative and anti-inflammatory nutrients were associated with an increase in serum bilirubin by 3% and reductions, respectively, in CRP, GGT, ALP, and white blood cell count by 7.4%, 3.3%, 5.2%, and 2.5%. | Colacino et al., 2014 [104] |
| Telomere length | NHANES 1999–2002, *n* 2093 with urinary Cd data, *n* 6796 with blood Cd plus Pb data. | Telomere shortening was associated with urinary and blood Cd levels but not blood Pb. | Zota et al., 2014 [105] |
| Telomere length | NHANES 1999–2002, *n* 7120 non-smokers, *n* 2296 smokers | A shorter telomere was associated with higher Cd exposure, CRP, trunk fat, and inactivity. A longer telomere was associated with retinyl stearate. | Patel et al., 2016 [106] |
| CRP and cardiovascular disease | NHANES 1999–2016 *n* 38,223 | CRP, triglycerides, total cholesterol, and white cell count were associated with elevated blood Cd levels. An increased risk of cardiovascular disease was associated blood Cd (OR 1.45). | Ma et al., 2022 [107] |
| Mortality | NHANES 2001–2010 Prospective, *n* 20,221, mean follow-up 9.1 years, *n* 2945 with diabetes | Risk of dying from all caused was increased by 49%, comparing blood Cd levels > 0.6 vs. < 0.24 µg/L. Cd, CRP, and 25(OH)D were associated with all-cause mortality among those with type 2 diabetes. | Liu et al., 2022 [108] |

NHANES, National Health and Nutrition Examination Survey; *n*, sample size; OR, odds ratio; GGT, γ-Glutamyl transferase; CRP, C-reactive protein; ALP, alkaline phosphatase.

## 4. Mitigation of the Cytotoxicity of Cadmium

Owing to its high toxicity and cumulative potential, minimizing the Cd contamination of the food chains and reducing Cd levels in food crops to the lowest achievable levels are essentially preventive public measures. Here, we discuss the frontline cellular stress response that may be a complementary measure to mitigate harmful effects of inevitable exposure to such a toxicant as Cd.

### 4.1. Heme Oxygenase-1 and Heme Oxygenase-2 (HO-1, HO-2)

HO-1 and HO-2 are enzymes involved in the degradation of heme to retrieve Fe for reuse by cells and to generate cytoprotective molecules, carbon monoxide (CO) and biliverdin IXα from which bilirubin is rapidly generated [109–111]. The economy of Fe utilization requires the salvaging of Fe, so the bulk of Fe released by the action of HO-1 and HO-2 is reutilized in the synthesis of hemoproteins, such as nitric oxide synthase, various enzymes of the mitochondrial respiratory chain, and the cytochrome P450 super family [112]. In every nucleated cell of the body, heme degradation and de novo biosynthe-

sis of heme are indispensable and simultaneous induction of MT and HO-1 occurs in most nucleated cells of the body in response to Cd exposure [32,109,110,113].

### 4.2. Products of the Physiologic Heme Degrdation

#### 4.2.1. Bilirubin

Serum bilirubin, a product of normal heme degradation and the catalytic activity of biliverdin XI-$\alpha$ reductase, contributes mostly to the total antioxidant capacity of blood plasma [114–116]. Due to its lipophilic properties, bilirubin is a lipid peroxidation chain breaker that protects lipids from oxidation more effectively than the water-soluble antioxidants, such as glutathione [115,116]. The ability of bilirubin to inhibit the oxidation of low-density lipoprotein accounts for the association observed between higher total serum bilirubin levels and lower risks of metabolic syndrome and non-alcoholic liver disease [117]. Of note, recent experimental data show that Cd-activated HO-1 gene and heme degradation did not result in formation of bilirubin [118]. A further discussion is in Section 5.

#### 4.2.2. Carbon Monoxide

Synthetic carbon monoxide-releasing molecules (CORM) were used to study effects of CO on mitochondrial biogenesis [119–121]. In high doses, CO has anti-inflammatory, anti-apoptotic, and vasodilatory effects and is cardioprotective. In low levels achievable through induction of HO-1 expression, CO increases the generation of reactive oxygen species (ROS) by the mitochondria, presumably through the inactivation of cytochrome C oxidase (COX) [119]. The elevated ROS then activates the PI3K/AKT signaling pathway, causing the inhibition of glycogen synthase kinase 3 β (GSK3β) and activation of the nuclear factor erythroid 2-related factor 2 (Nrf2) [122]. CO, p62, and NAD(P)H dehydrogenase quinone 1 (NQO1) are all required for the biogenesis of mitochondria and the removal of mitochondria with severe damage [122,123]. Mitochondrial ROS production is a mechanism that cells use to increase their capacity to adapt to stress [124,125]. Thus, HO-1 induction represents an important cellular stress response mechanism. The repression of this stress response gene is equally important to sustain the cellular redox state.

### 4.3. Role of HO-1, HO-2, and PFKFB4 in the Homeostasis of Blood Glucose

HO-1 and HO-2 are products of two different genes [126]. The promoter of the human HO-1 gene is unique because it contains the GT repeats, not found in rodent or murine species [109–111]. The genetic polymorphisms, such as long GT repeats, are associated with an elevated risk for various diseases, type 2 diabetes included [127,128].

Cellular expression of HO-1 is regulated by the transcription factor, including CLOCK, Bmal, and Per, that work together to generate day–night cyclical expression of the genes involved in energy metabolism [129–132]. Disruption of the diurnal cycle caused obesity in mice [133]. Expression of the HO-1 gene is controlled also by heme (its own substrate), the levels of glucose, oxygen, and shear stress [109,110,134,135].

The catalytic domains of HO-1 and HO-2 are highly homologous, sharing 93% of their amino acid sequences. HO-2, however, contains an additional domain, which has Cys-Pro dipeptide motifs that allows binding of heme and interacting with other proteins that include Rev-erbα, a heme sensor that coordinates metabolic and circadian pathways [136–138].

In addition to heme degradation activity, HO-2 has a regulatory role that was unraveled from obese and diabetic mice lacking HO-2 expression. HO-2 deficiency in mice caused neither lethality nor infertility, and HO-2 deficient mice underwent normal development to adulthood when they display the symptomatic spectrum of human type-2 diabetes, hyperglycemia, increased fat deposition, insulin resistance, and hypertension with aging [139–141]. The normal development and normal fecundity in the absence of HO-2 expression suggested that HO-1 could compensate for the heme-degradation activity of HO-2. However, HO-1 did not compensate for the anti-diabetogenicity and anti-obesity of HO-2.

In a protein microarray study, HO-2 was linked to the glycolytic pathway through its interaction with 6-phosphofructo-2-kinase/fructose-2,6-bisphosphatase 4 (PFKFB4) [142].

In liver, PFKFB4 is the key regulator of glycolysis [143], and a lack of HO-2 expression causes persistent hyperglycemia due to an impaired ability to suppress glucose production. Cd may mimic this effect of HO-2 deficiency, thereby causing hyperglycemia. Both HO-1 and HO-2 are required to prevent a fall of blood glucose during fasting or a rise in blood glucose in a post-absorptive period. HO-2 expression ensures PFKFB4 expression.

As Figure 3a,b indicate, the homeostasis of blood glucose requires coordinated activation and repression of HO-1, HO-2, and PFKFB4. Failure in any of these (HO-1, HO-2 and PFKFB4) causes hyperglycemic and obese phenotypes.

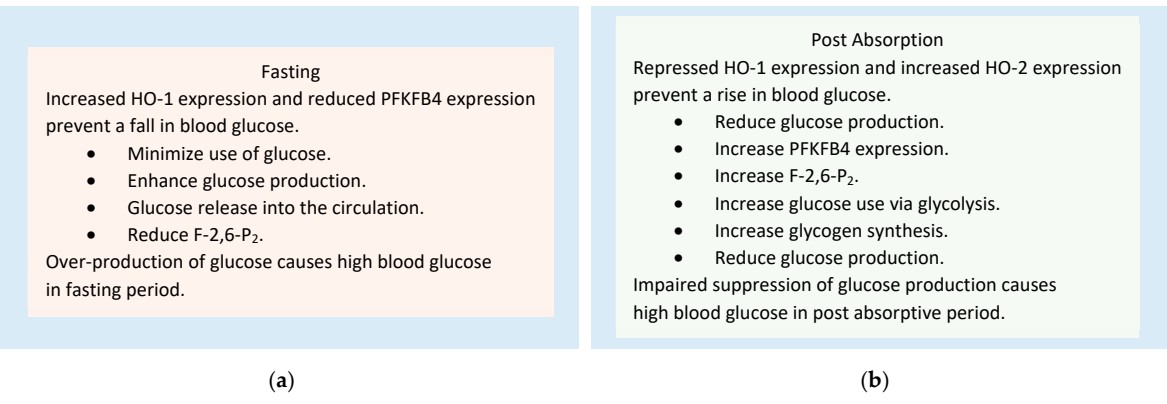

(**a**) (**b**)

**Figure 3.** Co-ordinated expression of HO-1, HO-2, and PFKFB4. (**a**) Changes in expression of HO-1 and PFKFB4 in the fasting state.; (**b**) Changes in expression of HO-1, HO-2, and PFKFB4 in the post absorptive period. Abbreviations: PFKFB4, 6-phosphofructo-2-kinase/fructose-2,6-biphosphatase 4; F-2,6-$P_2$, fructose 2,6-biphosphate.

In the liver of wild-type mice, lowered glycolysis with enhanced gluconeogenesis could be achieved in fasting state by HO-1 up-regulation plus PFKFB4 down-regulation. In the post-absorptive state, high glycolysis with suppressed gluconeogenesis could be achieved by HO-1 down-regulation plus HO-2 and PFKFB4 up-regulation. HO-1 protein expression levels in the liver of HO-2 knockout mice fell by 35–40% [144]. A possible consequence of a reduction in expression levels of HO-1 is increased susceptibility to oxidative damage. However, such repression of the HO-1 gene expression is an essential metabolic adaptation to safeguard the cellular redox state. This is achieved by utilizing NADPH ($H^+$) for regenerating reduced glutathione (GSH) rather than for heme catabolism [142]. GSH recycling is a mechanism for maintaining cellular redox state. It is central to normal protein folding and cell function (see Figure 4a in Section 4.3).

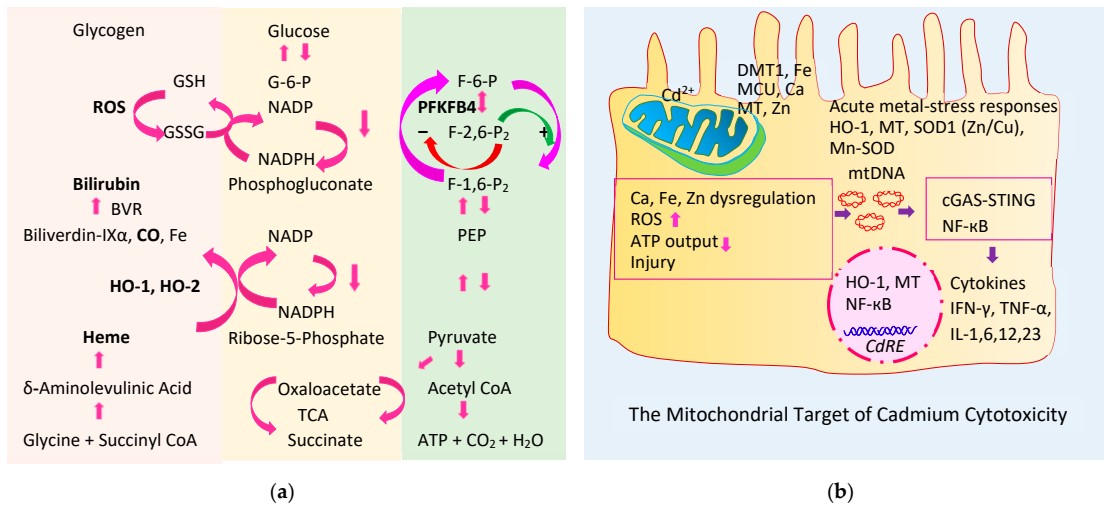

(**a**) (**b**)

**Figure 4.** Heme-glucose crosstalk and mitochondrial target of cadmium. (**a**) Heme degradation catalyzed by HO-1 and HO-2 utilizes NADPH from pentose phosphate pathway. (**b**) Cadmium-induced

kidney tubular cell death. Cd uses the Zn carrier metallothionein (MT) and transporters of Ca and Fe, mitochondrial calcium uniporter (MCU) and the divalent metal transporter 1 (DMT1) to reach the mitochondrial inner membrane [145]. There, Cd reduces ATP output and promotes reactive oxygen species (ROS) formation. Extensive damage causes a release of mitochondrial DNA (mtDNA). The DNA-sensing mechanism (cGAS-STING) and nuclear factor-kappaB (NF-κB) signaling pathways are activated, proinflammatory cytokines are released, and cell death ensues. Abbreviation: BVR, biliverdin reductase; PEP, phosphoenolpyruvate; cGAS, cyclic GMP-AMP synthase; STING, stimulator of interferon genes; *CdRE*, Cadmium response element.

### 4.4. Exogenous HO-1 Inducers

Several therapeutic drugs, such as statins (lipid lowering agents), rosiglitazone (anti-diabetic drug), aspirin (anti-inflammatory drug), paclitaxel and rapamycin (anti-cancer drugs), have been shown to induce HO-1 expression. The therapeutic efficacy of these drugs may be attributable, at least in part, to HO-1 induction [116,117].

A wide range of antioxidants from plant foods, such as curcumin, quercetin, tert-butylhydroquinone, and caffeic acid phenethyl ester, are HO-1 inducers, as are catechin (in green tea), $\alpha$-lipoic acid (in broccoli, spinach), resveratrol (in red wine, grapes), carnosol, sul-foraphane (cruciferous vegetable), coffee diterpenes cafestol, and kahweol [138–140,146–148]. Beneficial effects of consumption of these antioxidants could thus be mediated in part through the induction of HO-1 expression.

Diet high in anti-oxidative and anti-inflammatory nutrients was associated with increased serum bilirubin levels and reduced oxidative stress and systemic inflammation [104]. Green tea consumed in usual amounts was found to increase HO-1 expression [149–151]. One of the trials included only non-smoking diabetic subjects who had no history of metabolic complications and did not take regular food supplements [150]. Among 43 subjects, 23 had the long GT repeats (GT repeats $\geq$ 25; L/L genotype) type of the HO-1 promoter and another 20 had short GT repeats (GT repeats < 25; S/S genotype). According to Western blotting and the comet assay, HO-1 protein levels in circulating lymphocytes were increased by 40%, while the level of the DNA repair enzyme 8-oxoguanine glycosy-lase (hOGG1) was increased 50% with DNA damage being reduced by 15%. Green tea consumption increased HO-1 protein levels in lymphocytes in both L/L and S/S genotype groups, although the S/S group showed higher HO-1 protein levels at baseline, compared to the L/L group. This trial showed that green tea consumption may reduce cellular DNA damage through induced expression of HO-1.

## 5. Different HO-1 Gene Activation Mechanisms

All nucleated cells of the body have the capacity to take up Cd from the circulation and they must synthesize their own heme for their own use. A de novo biosynthesis of heme is a requisite for cellular response to stressors, and this has been demonstrated for Cd as a stressor [117]. Current evidence suggests that Cd induces the expression of HO-1 by mechanisms different from those used by endogenous (physiologic) HO-1 activators (Section 4.3) and prostaglandin D2 (PGD$_2$) [152].

PGD$_2$ is a major cyclooxygenase mediator, synthesized by activated mast cells and other immune cells and is implicated in allergic disorders [153]. In a study of a cell culture model of human retinal epithelial cells, PGD$_2$ was found to activate the HO-1-gene promoter through D-prostanoid 2 (DP$_2$) receptor in an enhancer manner [152]. In contrast, Cd activates the HO-1 promoter via the Cd response element (*CdRE*, TGCTAGATTTT) and Maf recognition antioxidant response element (MARE, GCTGAGTRTGACNNNGC), also known as stress response element (*StRE*) [113]. Cd also suppresses lysosomal degradation of Nrf2 [154] and causes nuclear export of the HO-1 gene repressor Bach1, which allows transactivation of the HO-1 gene by the Nrf2/small Maf complex [155]

Cd-induced expression of the HO-1 increases intracellular concentration of heme, a stimulator of gluconeogenesis and known cause of hyperglycemia. This may explain

hyperglycemic state induced by Cd. However, Cd-induced expression of HO-1 does not result in the formation of bilirubin [118]. The reason for this phenomenon remains unclear, but it may explain an increased cellular oxidative stress through lowering levels of the anti-oxidative molecule, bilirubin.

## 6. Conclusions

This narrative review focused on adverse metabolic outcomes of chronic exposure to Cd. Epidemiologic data indicate that environmental exposure to low levels of Cd increases the risk of type 2 diabetes by several mechanisms that may include oxidative stress, inflammation, adipose tissue dysfunction, increased insulin resistance, and a dysregulated cellular intermediary metabolism. Higher levels of Cd accumulation in adipose tissues are associated with lower BMI and increased insulin resistance. A statistically significant inverse association between Cd exposure and obesity is universally observed in both children and adults. Thus, Cd-induced type 2 diabetes is independent of adiposity.

The cellular stress response mechanisms involving the catabolism of heme, mediated by HO-1 and HO-2, may mitigate the cytotoxicity of Cd. The products of their physiologic heme degradation, bilirubin, and carbon monoxide have antioxidative, anti-inflammatory, and anti-apoptotic properties. Exogenous HO-1 inducers may raise the quantities of these protective molecules, and thus could be a complementary measure to mitigate the cytotoxicity of Cd. However, strategies that minimize Cd entry into the food chains remain essential preventive public measures.

Knowledge gained from the phenotypic analyses of HO-2 deficient mice that display the symptomatic spectrum of human type 2 diabetes have shown that HO-1, HO-2, and PFKFB4, the key regulators of glycolysis, are required to prevent hyperglycemia and an obese phenotype. Repression of the HO-1 gene expression is an essential metabolic adaptation of equal importance to safeguard the cellular redox state.

Cd induces the expression of HO-1 by mechanisms different from those of physiologic HO-1 gene activators, and consequently Cd-induced HO-1 expression does not produce bilirubin as a product. This may represent one of the cytotoxic mechanisms of Cd, which is in addition to a hyperglycemic phenotype.

**Author Contributions:** S.S. conceptualized the review and prepared an initial draft with G.C.G. and D.A.V. providing logical data interpretation. G.C.G. and D.A.V. reviewed and edited the draft manuscript. All authors have read and agreed to the published version of the manuscript.

**Funding:** This work received no external funding.

**Institutional Review Board Statement:** Not applicable.

**Informed Consent Statement:** Not applicable.

**Data Availability Statement:** All data are contained within this article.

**Acknowledgments:** This review is dedicated to the late Michael R. Moore, who was Director of the National Research Centre for Environmental Toxicology (EnTox), University of Queensland, between 1994 and 2009. He was instrumental in establishing toxicology research on heavy metals in Australia, and he was an inspiration to all who worked in this field. S.S. thanks Professor Shigeki Shibahara for his patience, support, and guidance on HO-1 and HO-2 research undertaken by the author at Tohoku University, Sendai, Japan. This work was supported with resources from the Kidney Disease Research Collaborative and the Department of Nephrology, Princess Alexandra Hospital.

**Conflicts of Interest:** The authors declare no conflict of interest.

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
