# Peer review of "Mitigation of Cadmium Toxicity through Modulation of the Frontline Cellular Stress Response"

_stresses, doi:10.3390/stresses2030025_

Round 1

Reviewer 1 Report

The manuscript is interesting, but some improvements maybe needed.

1-     In the abstract, after the objective, the remaining statements are not suitable for the abstract (lines 24 to 30). Authors may use the statements in the conclusion part of the main text to give readers a summary of their findings/conclusion.

2-     It is not clear why “anti-obesity” is listed in keywords

3-     Lines 94 to 97, “However, 94 various species of mixed metal complexes such as Cd3Cu3ZnMT, Cd4CuZn2MT and 95 Cd6CuMT are formed in vivo, with the molar contents of Cd depent on levels of Cd expo-96 sure [29]”. Please elaborate on the consequences of the formation of these mixed metal complex

4-     Lines 101to 104, “Although the formation of CdMT complexes prevent acute cytotoxicity, it pro-101 vides also an opportunity for long-term toxicity because Cd2+ ions can be released under 102 certain conditions leading to an increased synthesis of nitric oxide (NO) that liberates the 103 Cd2+ ions previously bound to MT [30-32].”, replace the word opportunity” with a more suitable term such as “risk”

5-     Line 113, “The estimated half-life of blood Cd ranged between 75 and 128 days “, is this referring to biological half life?

6-     Lines 110- 111, “Plasma/serum Cd is more predictive of its toxicity than those in erythrocytes.”, this statement is confusing

7-     Line 122, “………….epithelial cells of the kidneys, discussed in Section 2.4 below”, please remove the word “below” as the section number is enough clear to address the text position.

8-     Line  128, “ glomerular filtration n membrane”, what is the letter “n”?

9-     In table one, after n, please add “=”. Same for table 2

10- Line 180, “Increases in the risks of prediabetes and diabetes among in NHANES 1988-1994 ………”, remove either “among” or “in”

11- Please elaborate on how statements in lines 280 - 288, are related to Cd which is the focus of this manuscript

12- It is said in lines 355-357 that: “Current evidence suggests that Cd induces expression of HO-1 by mechanisms different from those used by endogenous (physiologic) HO-1 activators, as listed above, and 356 prostaglandin D2 [135]”. Then it is hard to justify the relevance of statements in subheadings 4.2 and 4.3 to the objective of this study. Authors may want to rewrite these subheadings to be more linked to their objective or remove them. It can be understood that the authors have tried to “discuss the frontline cellular stress response that may be a complementary measure to mitigate harmful effects of inevitable exposure to such a toxicant as Cd”, but they need to keep the flow of the write-up and find connections between ideas

Author Response

Reviewer 1.

We thank the Reviewer for comments and for the suggestions to improve our paper. We have carefully considered the issues raised, and made an effort to fully address them.  We provide below point-by-point response to issues/concerns raised. Changes made to the text in a manuscript are in blue letters as are additional references.

The manuscript is interesting, but some improvements maybe needed.

Point 1. In the abstract, after the objective, the remaining statements are not suitable for the abstract (lines 24 to 30). Authors may use the statements in the conclusion part of the main text to give readers a summary of their findings/conclusion.

Response:

  • Referred statements has been replaced by a succinct summary of key findings from reviewing the literature reports (lines 22-30).

Point 2. It is not clear why “anti-obesity” is listed in keywords

Response:

  • “Anti-obesity” has been deleted.

Point 3. Lines 94 to 97, “However, various species of mixed metal complexes such as Cd3Cu3ZnMT, Cd4CuZn2MT and Cd6CuMT are formed in vivo, with the molar contents of Cd depent on levels of Cd exposure [29]”. Please elaborate on the consequences of the formation of these mixed metal complex.

Response:

  • We have now explained the consequences of the Cu/Zn/Cd/MT complexes (lines 98-103) together with six additional references (ref. 30-36).

Point 4. Lines 101 to 104, “Although the formation of CdMT complexes prevent acute cytotoxicity, it provides also an opportunity for long-term toxicity because Cd2+ ions can be released under certain conditions leading to an increased synthesis of nitric oxide (NO) that liberates the Cd2+ ions previously bound to MT [30-32].”, replace the word opportunity” with a more suitable term such as “risk”

Response:

  • The sentence has been reworded to read, “it may increase the risk of long-term toxicity”.

Point 5. Line 113, “The estimated half-life of blood Cd ranged between 75 and 128 days “, is this referring to biological half life?

Response:

  • Correction has been undertaken to indicate that it is a biological half-life of blood Cd.

Point 6. Lines 110- 111, “Plasma/serum Cd is more predictive of its toxicity than those in erythrocytes.”, this statement is confusing

Response:

  • The referred sentence has bene deleted.

Point 7. Line 122, “………….epithelial cells of the kidneys, discussed in Section 2.4 below”, please remove the word “below” as the section number is enough clear to address the text position.

Response:

  • The word “below” has been removed.

Point 8. Line 128, “glomerular filtration n membrane”, what is the letter “n”?

Response:

  • A typo error has now been removed.

Point 9. In table one, after n, please add “=”. Same for table 2

Response: In accordance with the journal style, we have used an italicized “n” to denote a sample size.

Point 10. Line 180, “Increases in the risks of prediabetes and diabetes among in NHANES 1988-1994 ………”, remove either “among” or “in”

Response:

  • A typo error “in” has been removed.

Point 11. Please elaborate on how statements in lines 280 - 288, are related to Cd which is the focus of this manuscript

Response

  • Referred statements have been moved to subheading 4.3 “Role of HO-1, HO-2 and PFKFB4 in the Homeostasis of Blood Glucose” (line 330). Please see also response to point 12.

Point 12. It is said in lines 355-357 that: “Current evidence suggests that Cd induces expression of HO-1 by mechanisms different from those used by endogenous (physiologic) HO-1 activators, as listed above, and prostaglandin D2 [135]”. Then it is hard to justify the relevance of statements in subheadings 4.2 and 4.3 to the objective of this study. Authors may want to rewrite these subheadings to be more linked to their objective or remove them. It can be understood that the authors have tried to “discuss the frontline cellular stress response that may be a complementary measure to mitigate harmful effects of inevitable exposure to such a toxicant as Cd”, but they need to keep the flow of the write-up and find connections between ideas

Response: To keep the flow of ideas, as advised, we have made the following changes to subheadings and the text.

  • A word “physiologic” has been added to a title of subheading 4.2, which now read, “Products of the Physiologic Degradation of Heme”.
  • Subheading 4.3 has been changed to “Role of HO-1, HO-2 and PFKFB4 in the Homeostasis of Blood Glucose”.
  • The statements previously appearing in lines 280 – 288 have been moved to a new subheading 4.3.
  • Exogenous HO-1 Inducers are now under subheading 4.4.
  • Different HO-1 Gene Activation Mechanisms are in a new Section 5, where the mechanisms by which Cd activates the HO-1 gene is described in comparison with those of prostaglandin D2, a representative of physiologic inducer of the HO-1 gene (lines 421-435, 153-155). 

Reviewer 2 Report

This manuscript is a review of mechanisms of cadmium toxicity especially with regard to its manifestation in dysregulation of glucose metabolism and fat cells and onset of type 2 diabetes. It is well written and requires only minor revision.

1. Abstract line 16, introduction lines 49-50: The authors suggest that there is no cellular excretory mechanism for Cd. The authors should consider that cells see Cd as a Zn homologue in many ways including transport into cells. When sufficient zinc is accumulated in cells, the zinc transporter activity is decreased. However, if cadmium exposure/circulating cadmium concentrations are high, cadmium will be imported along with the zinc and increase the total cell Zn + Cd concentration, suppressing expression of passive transporters that would otherwise be a mechanism of Zn or Cd efflux. Have the authors not considered zinc transporter proteins as inadvertent mechanisms for influx/efflux of cellular Cd as well, until the total Cd + Zn suppresses transporter activity? The authors should examine the following and consider revising their comments on lack of a transport mechanism for Cd or make a strong case against zinc transporters as inadvertent cadmium transporters:

Costello, L.C.; Levy, B.A.; Desouki, M.M.; Zou, J.; Bagasra, O.; Johnson, L.A.; Hanna, N.; Franklin, R.B. Decreased zinc and downregulation of ZIP3 zinc uptake transporter in the development of pancreatic adenocarcinoma. Cancer Biol. Ther. 2011, 12, 297–303.

Franklin, R.B.; Zou, J.; Costello, L.C. The cytotoxic role of RREB1, ZIP3 zinc transporter, and zinc in human pancreatic adenocarcinoma. Cancer Biol. Ther. 2014, 15, 1431–1437.

Li, M.; Zhang, Y.; Liu, Z.; Bharadwaj, U.;Wang, H.;Wang, X.; Zhang, S.; Liuzzi, J.P.; Chang, S.M.; Cousins, R.J.; et al. Aberrant expression of zinc transporter ZIP4 (SLC39A4) significantly contributes to human pancreatic cancer pathogenesis and progression. Proc. Natl. Acad. Sci. USA 2007, 104, 18636–18641.

Costello, L.C.; Levy, B.A.; Desouki, M.M.; Zou, J.; Bagasra, O.; Johnson, L.A.; Hanna, N.; Franklin, R.B. Decreased zinc and downregulation of ZIP3 zinc uptake transporter in the development of pancreatic adenocarcinoma. Cancer Biol. Ther. 2011, 12, 297–303.

Franklin, R.B.; Zou, J.; Costello, L.C. The cytotoxic role of RREB1, ZIP3 zinc transporter, and zinc in human pancreatic adenocarcinoma. Cancer Biol. Ther. 2014, 15, 1431–1437.

Li, M.; Zhang, Y.; Liu, Z.; Bharadwaj, U.;Wang, H.;Wang, X.; Zhang, S.; Liuzzi, J.P.; Chang, S.M.; Cousins, R.J.; et al. Aberrant expression of zinc transporter ZIP4 (SLC39A4) significantly contributes to human pancreatic cancer pathogenesis and progression. Proc. Natl. Acad. Sci. USA 2007, 104, 18636–18641.

2. Page 3 lines 86-88: Authors suggest that kidney tubular cells are well equipped to reabsorb filtered proteins for reabsorption. Retinol Binding Protein (RBP), a relatively small protein, circulates bound to transthyretin (TTR). One of the well known consequences of cadmium toxicity is disruption of the RBP-TTR complex with subsequent filtration of RBP by the kidneys and excretion in the urine. The authors should consider this and possibly modify their statement.

3. Lines 99-100: Here and in other places where hepatic absorption and accumulation of cadmium is described mention cadmium hepatotoxicity and possible release of Cd-MT as a consequence of cell death, but I did not see any description of the metabolism of cadmium by conjugation with glutathione (or dithioerythritol) and excretion via the gall bladder in the bile. The authors should examine the following manuscripts and comment on this:

Graf P, Sies H. Hepatic uptake of cadmium and its biliary release as affected by dithioerythritol and glutathione. Biochem Pharmacol. 1984 Feb 15;33(4):639-43. doi: 10.1016/0006-2952(84)90320-4. PMID: 6704180.

Elinder CG, Kjellstöm T, Lind B, Molander ML, Silander T. Cadmium concentrations in human liver, blood, and bile: comparison with a metabolic model. Environ Res. 1978 Oct;17(2):236-41. doi: 10.1016/0013-9351(78)90025-7. PMID: 318516.

Ren L, Qi K, Zhang L, Bai Z, Ren C, Xu X, Zhang Z, Li X. Glutathione Might Attenuate Cadmium-Induced Liver Oxidative Stress and Hepatic Stellate Cell Activation. Biol Trace Elem Res. 2019 Oct;191(2):443-452. doi: 10.1007/s12011-019-1641-x. Epub 2019 Feb 4. PMID: 30715683.

Author Response

Reviewer 2

We thank the Reviewer for insightful comments and useful suggestions. We have studied all references and issues raised. Changes made to the text in a manuscript are in blue letters as are additional references. Our point-by-point responses are as detailed below

This manuscript is a review of mechanisms of cadmium toxicity especially with regard to its manifestation in dysregulation of glucose metabolism and fat cells and onset of type 2 diabetes. It is well written and requires only minor revision.

  1. Abstract line 16, introduction lines 49-50: The authors suggest that there is no cellular excretory mechanism for Cd. The authors should consider that cells see Cd as a Zn homologue in many ways including transport into cells. When sufficient zinc is accumulated in cells, the zinc transporter activity is decreased. However, if cadmium exposure/circulating cadmium concentrations are high, cadmium will be imported along with the zinc and increase the total cell Zn + Cd concentration, suppressing expression of passive transporters that would otherwise be a mechanism of Zn or Cd efflux. Have the authors not considered zinc transporter proteins as inadvertent mechanisms for influx/efflux of cellular Cd as well, until the total Cd + Zn suppresses transporter activity? The authors should examine the following and consider revising their comments on lack of a transport mechanism for Cd or make a strong case against zinc transporters as inadvertent cadmium transporters:

Costello, L.C.; Levy, B.A.; Desouki, M.M.; Zou, J.; Bagasra, O.; Johnson, L.A.; Hanna, N.; Franklin, R.B. Decreased zinc and downregulation of ZIP3 zinc uptake transporter in the development of pancreatic adenocarcinoma. Cancer Biol. Ther. 2011, 12, 297–303.

Franklin, R.B.; Zou, J.; Costello, L.C. The cytotoxic role of RREB1, ZIP3 zinc transporter, and zinc in human pancreatic adenocarcinoma. Cancer Biol. Ther. 2014, 15, 1431–1437.

Li, M.; Zhang, Y.; Liu, Z.; Bharadwaj, U.;Wang, H.;Wang, X.; Zhang, S.; Liuzzi, J.P.; Chang, S.M.; Cousins, R.J.; et al. Aberrant expression of zinc transporter ZIP4 (SLC39A4) significantly contributes to human pancreatic cancer pathogenesis and progression. Proc. Natl. Acad. Sci. USA 2007, 104, 18636–18641.

Costello, L.C.; Levy, B.A.; Desouki, M.M.; Zou, J.; Bagasra, O.; Johnson, L.A.; Hanna, N.; Franklin, R.B. Decreased zinc and downregulation of ZIP3 zinc uptake transporter in the development of pancreatic adenocarcinoma. Cancer Biol. Ther. 2011, 12, 297–303.

Franklin, R.B.; Zou, J.; Costello, L.C. The cytotoxic role of RREB1, ZIP3 zinc transporter, and zinc in human pancreatic adenocarcinoma. Cancer Biol. Ther. 2014, 15, 1431–1437.

Li, M.; Zhang, Y.; Liu, Z.; Bharadwaj, U.;Wang, H.;Wang, X.; Zhang, S.; Liuzzi, J.P.; Chang, S.M.; Cousins, R.J.; et al. Aberrant expression of zinc transporter ZIP4 (SLC39A4) significantly contributes to human pancreatic cancer pathogenesis and progression. Proc. Natl. Acad. Sci. USA 2007, 104, 18636–18641.

Response:

  • In abstract, our statement “no excretory mechanisms” has been changed to “Due to an extremely slow elimination rate, most Cd is retained by cells” (line.
  • We acknowledge the suggestion to make a strong case against zinc transporters as inadvertent Cd transporters. This has been addressed by the addition a new subheading paragraphs detailing experimental data showing involvement of zinc influx transporter, ZIP8 and zinc efflux transporter, ZnT1 in cellular Cd uptake, cytotoxicity and tolerance (line 16).

  1. Page 3 lines 86-88: Authors suggest that kidney tubular cells are well equipped to reabsorb filtered proteins for reuse. Retinol Binding Protein (RBP), a relatively small protein, circulates bound to transthyretin (TTR). One of the well known consequences of cadmium toxicity is disruption of the RBP-TTR complex with subsequent filtration of RBP by the kidneys and excretion in the urine. The authors should consider this and possibly modify their statement.

Response:

  • A discussion on RBP and its excretion due to a reduction in tubular reabsorption caused by Cd-induced tubular injury plus nephron loss has been added together with references (lines 159-167). A new Figure 2 has been inserted to show appearance of filtered proteins, RBP and β2MG in urine, resulting from reduced reabsorption and nephron loss.

  1. Lines 99-100: Here and in other places where hepatic absorption and accumulation of cadmium is described mention cadmium hepatotoxicity and possible release of Cd-MT as a consequence of cell death, but I did not see any description of the metabolism of cadmium by conjugation with glutathione (or dithioerythritol) and excretion via the gall bladder in the bile. The authors should examine the following manuscripts and comment on this:

Graf P, Sies H. Hepatic uptake of cadmium and its biliary release as affected by dithioerythritol and glutathione. Biochem Pharmacol. 1984 Feb 15;33(4):639-43.

Elinder CG, Kjellstöm T, Lind B, Molander ML, Silander T. Cadmium concentrations in human liver, blood, and bile: comparison with a metabolic model. Environ Res. 1978 Oct;17(2):236-41.

Ren L, Qi K, Zhang L, Bai Z, Ren C, Xu X, Zhang Z, Li X. Glutathione Might Attenuate Cadmium-Induced Liver Oxidative Stress and Hepatic Stellate Cell Activation. Biol Trace Elem Res. 2019 Oct;191(2):443-452.

Response:

  • Biliary excretion of Cd and potential protective effects of glutathione are now discussed under a new subheading 2.4. Roles for Zinc Transporters in Biliary Excretion and Cytotoxicity of Cadmium (lines 130-152). All references suggested are cited.
